# Gene Therapy of Sphingolipid Metabolic Disorders

**DOI:** 10.3390/ijms24043627

**Published:** 2023-02-11

**Authors:** Alisa A. Shaimardanova, Valeriya V. Solovyeva, Shaza S. Issa, Albert A. Rizvanov

**Affiliations:** 1Institute of Fundamental Medicine and Biology, Kazan Federal University, 420008 Kazan, Russia; 2Department of Genetics and Biotechnology, St. Petersburg State University, 199034 St. Petersburg, Russia

**Keywords:** gene therapy, AAV, LV, sphingolipidosis, sphingolipid metabolic disorders, cell therapy, sphingolipid lysosomal storage diseases

## Abstract

Sphingolipidoses are defined as a group of rare hereditary diseases resulting from mutations in the genes encoding lysosomal enzymes. This group of lysosomal storage diseases includes more than 10 genetic disorders, including GM1-gangliosidosis, Tay–Sachs disease, Sandhoff disease, the AB variant of GM2-gangliosidosis, Fabry disease, Gaucher disease, metachromatic leukodystrophy, Krabbe disease, Niemann–Pick disease, Farber disease, etc. Enzyme deficiency results in accumulation of sphingolipids in various cell types, and the nervous system is also usually affected. There are currently no known effective methods for the treatment of sphingolipidoses; however, gene therapy seems to be a promising therapeutic variant for this group of diseases. In this review, we discuss gene therapy approaches for sphingolipidoses that are currently being investigated in clinical trials, among which adeno-associated viral vector-based approaches and transplantation of hematopoietic stem cells genetically modified with lentiviral vectors seem to be the most effective.

## 1. Introduction

Sphingolipidoses are defined as rare hereditary diseases belonging to the group of lysosomal storage diseases (LSD). LSDs are a group of approximately 50 genetic disorders caused by mutations in the genes encoding enzymes that are involved in cellular degradation and transport of lipids and other macromolecules. Abnormal accumulation of lipids and other macromolecules in lysosomes results in death of affected cells. Although clinical manifestations of different LSDs vary greatly, over half of LSDs are associated with nervous system degeneration symptoms [1,2,3,4,5].

Sphingolipidoses include more than 10 diseases, such as GM1-gangliosidosis, Tay–Sachs disease, Sandhoff disease, the AB variant of GM2-gangliosidosis, Fabry disease, Gaucher disease, metachromatic leukodystrophy, Krabbe disease, Niemann–Pick disease type A and B, Farber disease, combined saposin (Sap) deficiency, SapA deficiency (atypical Krabbe disease), SapB deficiency (atypical metachromatic leukodystrophy), and SapC deficiency (atypical Gaucher disease). Table 1 provides brief descriptions of these diseases. In general, sphingolipidoses are inherited in an autosomal recessive (AR) manner, except for Fabry disease, which is an X-linked disease (XL). These diseases result from mutations either in the genes encoding lysosomal enzymes or those encoding activator proteins of their substrates (sphingolipids), causing toxic accumulation of sphingolipids in the lysosomes of cells in various organs [6].

To date, sphingolipidoses disorders remain incurable, with extremely limited therapeutic variants. However, in recent decades, there has been an active search to find new effective therapeutic approaches for this type of disease. Currently used approaches include bone marrow transplantation (BMT) [38,39,40,41], enzyme replacement therapy (ERT) [42,43], substrate reduction therapy (SRT) [8,44,45,46], chaperone therapy [47,48,49,50], and supportive care [51].

BMT is believed to be able to correct metabolic defects and lead to improved metabolism in most patients. Hematopoietic stem cells (HSCs) are known to be a source of lysosomal enzymes, but their expression level in native HSCs is sometimes insufficient. However, genetically modified HSCs have shown more promising outcomes [52], which will be discussed in detail below. SRT involves using small molecules, which are usually able to overcome the BBB, in order to slow the rate of sphingolipid biosynthesis. However, SRT cannot prevent neurological deterioration in patients [1,8,44,53,54]. For diseases caused by missense mutations that affect proper protein folding and maturation without completely abolishing its catalytic activity, regulation of proteostasis may be useful. In this process, molecular chaperones, which are important for the functioning of lysosomal enzymes, are isolated. Such an approach can have many advantages, such as the possibility of drug repositioning and the use of small molecules capable of overcoming the BBB. Another innovative approach is translational readthrough. Nonsense mutations are caused by premature termination codons, which can lead to null phenotypes. Such mutations can be corrected by drugs that produce the readthrough of a premature termination codon, which can enable synthesis of a full-length protein [55,56]. All these methods often improve patients’ well-being and their quality of life; however, in many cases, they can be insufficient treatments. Accordingly, combined therapy that includes several treatment approaches provides the most desired results [57,58].

ERT is considered the current standard of care for sphingolipidoses, and most currently approved drugs are based on this type of treatment. However, most sphingolipidoses affect the central (CNS) and peripheral nervous systems, which is a limiting factor for the use of ERT [59,60]. Gene therapy seems to be a promising approach for sphingolipidose treatment. It has an important advantage over ERT, as ERT requires lifelong biweekly infusions, while gene drugs administration can be performed much less frequently, even as a single dose in some cases [61]. However, regardless of the choice of treatment approach, many studies highlight the need to start treatment before the onset of symptoms [52,62,63]. Therefore, timely diagnosis of these diseases is an extremely important requirement that is not met in all cases. Because sphingolipidoses are orphan diseases (some of which are extremely rare), they are not always correctly diagnosed, considering that the symptoms may be similar to those of other more common diseases, and an accurate diagnosis can be made only after applying various molecular genetic methods. Therefore, publishing scientific discoveries in this area, specifically providing information about ongoing clinical trials, is of a great importance. In this review, we discuss existing gene therapies and gene therapy methods that are undergoing clinical trials for the treatment of sphingolipidoses, along with the results of some previous preclinical studies.

## 2. Gene Therapy for Sphingolipidoses

The first clinical trial of a gene therapy approach was registered in 1988 (NCT00001234). The study included 120 patients with Gaucher and Fabry diseases with the aim of investigating the therapeutic effect of transplantation of autologous CD34^+^ HSCs (CD34^+^) genetically modified with a retroviral vector. This clinical trial is the only one to use a retroviral vector in sphingolipidoses. In 1996, C. Dunbar and D. Kohn published a detailed protocol for a clinical study involving 24 patients with Gaucher disease who were treated with a retroviral vector encoding the β-glucocerebrosidase 1 (*GBA1*) gene [63]. Later in 1998, along with other coauthors, the researchers published the results of three patients and showed that transduced cells had successfully engrafted and persisted in patients for at least 3 months after infusion. However, β-glucocerebrosidase 1 enzymatic activity did not increase in any patient after transduced cell infusion [64].

Further gene therapy approaches for sphingolipidoses have developed mainly in two directions, leading to the registration of clinical trials using either vectors based on adeno-associated viruses (AAV) or transplantation of CD34^+^ cells transduced with lentiviral vectors (LV). Due to their good safety profile and successful recent clinical trials that have led to the registration of several gene drugs, these strategies have been broadly developed.

According to the clinicaltrials.gov database, a total of 29 clinical trials aimed at studying gene therapy for sphingolipidoses have been registered to date (Table 2, Figure 1), among which 18 involve AAV-based vectors, 10 involve LV, and 1 involves the retroviral vector discussed above. A comparison of the two approaches (LV and AAV) shows that the use of LV began earlier (2010) than AAV (2014); however, in the last few years, the number of registered clinical trials using AAV has increased, in contrast to LV. Analysis of clinical trial databases did not reveal works aimed at studying gene therapy approaches for Niemann–Pick disease types A and B, Farber disease, the AB variant of GM2-gangliosidosis, or saposin deficiencies. This may be due to the fact that these diseases are rarer and less studied. For Niemann–Pick disease types A and B, there are ongoing gene therapy preclinical trials, and clinical studies may be registered in the near future [65].

## 3. Adeno-Associated Viruses in Gene Therapy

AAVs are well known to be the safest vectors for gene therapy, and many studies have shown their efficacy both in various animal models and human patients. To date, five AAV-based drugs have been approved. In 2012, the European Medicines Agency (EMA) approved the first AAV-based gene drug, Glybera, which is an AAV serotype 1 (AAV1). The drug was developed for the treatment of hereditary lipoprotein lipase deficiency. However, it turned out to be commercially unprofitable due to the rarity of the disease (1–2 cases per 1 million people), and its production was therefore discontinued [99,100,101]. Later in 2017, as part of a speeded process (priority review), the U.S. Food and Drug Administration (FDA) approved AAV2 drug Luxturna for the treatment of patients with hereditary retinal dystrophy caused by a biallelic gene mutation in the *RPE65* gene (Leber’s congenital amaurosis) [99,102]. Zolgensma, an AAV9-based drug for the treatment of spinal muscular atrophy, was approved by the FDA in 2019, followed by the EMA and several other countries [103,104]. In 2022, two additional novel AAV-based drugs, Roctavian and Hemgenix, were also approved. Roctavian, is an AAV5-based gene therapy that has been approved by the EMA for the treatment of severe hemophilia A, which is caused by a mutation in the gene encoding factor VIII (F8) [105], while Hemgenix, also an AAV5-based gene therapy, has been approved by the FDA for the treatment of patients with hemophilia B.

### 3.1. AAV-Mediated Gene Therapy for GM1-Gangliosidosis

GM1-gangliosidosis results from mutations in the β-galactosidase 1 (GLB1) gene, which, in turn, cause enzyme deficiency and accumulation of GM1-ganglioside in various tissues of patients, predominantly affecting the nervous system [7]. Currently, three types of AAV vectors encoding the *GLB1* gene are undergoing clinical trials: AAV9, AAVhu68, and AAVrh10. These serotypes were chosen for their ability to cross the blood–brain barrier (BBB) and effectively transduce nerve cells, as patients with GM1-gangliosidosis primarily suffer from progressive degeneration of neural tissue. AAV9 [106,107,108,109,110], and AAVrh10 [67,111,112] are often used in gene therapy for various diseases affecting the nervous system. In 2019, an AAV9-based gene drug under the name Zolgensma was registered for the treatment of spinal muscular atrophy. This drug has been shown to achieve successful transduction of motor neurons following intravenous administration, and single-dose administration resulted in the maintenance of motor activity in sick children [103,113]. AAVhu68, also a neurotropic vector, is one a new variant of AAV9. Studies have confirmed the high efficiency of gene transfer using this vector in the brain and spinal cord following administration into the cerebrospinal fluid [25,68,114,115,116].

Unfortunately, results of clinical studies of AAV9 (NCT03952637), AAVhu68 (NCT04713475) and AAVrh10 (NCT04273269) encoding the *GLB1* gene are not available yet, but results of their preclinical studies have been published. In preclinical studies, AAV9 encoding *GLB1* was administered intravenously to 4-week-old GM1-ganliosidosis model mice, resulting in widespread and sustained expression of *GLB1* up to 32 weeks, which led to a reduction in GM1-ganglioside accumulation, myelin deficiency, and damage of neurons in the area of the cerebral cortex. The decrease in GM1-ganglioside was also accompanied by a decrease in activated microglia [66].

An AAVrh10 vector encoding the wild-type human gene *GLB1* driven by cytomegalovirus (CMV) enhancer fused to a chicken β-actin promoter/rabbit β-globin intron was developed for clinical studies. The efficacy of analogs of this vector containing wild-type mouse or feline *GLB1* genes was studied in mouse and cat models of GM1-gangliosidosis. Various delivery routes to the CNS have been investigated. Mice were injected bilaterally into the thalamus (3.5 × 10^9^, 1.0 × 10^10^, 3.5 × 10^10^, 1.0 × 10^11^ viral genome copies (vg)/animal) or unilaterally into the lateral ventricle (3.5 × 10^10^, 1.0 × 10^11^, 3.5 × 10^11^ vg/animal). Direct intrathalamic injection resulted in dose-dependent toxicity at the two highest doses of 3.5 × 10^10^ and 1.0 × 10^11^ vg/animal, while unilateral injection into the lateral ventricle showed no toxicity and a wide distribution, along with a significant dose-dependent increase in GLB1 levels and a decrease in GM1-ganglioside in the brain after injections. Cats were administered 1.0 × 10^12^ vg/kg through different routes (intracisternal magna, intracerebroventricular, and intrathecal lumbar) providing widespread delivery of GLB1 to the CNS. However, intracisternal magna and intracerebroventricular routes of administration were preferable, as intrathecal lumbar administration did not lead to an increase in GLB1 activity in the brain. Toxicity and biodistribution of two doses (low 1.4 × 10^13^ vg/animal; high 5.4 × 10^13^ vg/animal) following administration in the cisternal magna space were analyzed in nonhuman primates. A wide distribution of GLB1 was detected without serious side effects [67].

Preclinical studies of AAVhu68 were performed in a mouse model of GM1-gangliosidosis. An AAVhu68 vector encoding codon-optimized human *GLB1* genes was developed. Interestingly, researchers compared the efficiency of several promoter types for this vector in vivo: (1) chicken beta actin with a CMV enhancer, (2) elongation factor 1 alpha, and (3) ubiquitin C (UbC). Remarkably, only the UbC vector resulted in a statistically significant increase in GLB1 enzymatic activity both in the brain and cerebrospinal fluid. Intracerebroventricular mice were injected with varying doses (4.4 × 10^9^, 1.3 × 10^10^, 4.4 × 10^10^, or 1.3 × 10^11^ vg). An almost full correction of the disease phenotype was achieved at a dose of 4.4 × 10^10^ vg (corresponding to 1.1 × 10^11^ vg/g brain mass). An increase in the enzyme activity was detected in the brain, along with a reduction in neuronal damage, prevention of neurological symptoms, and increased survival [68].

### 3.2. AAV-Mediated Gene Therapy for GM2-Gangliosidoses

GM2-gangliosidoses result from deficiency of β-hexosaminidase A (HEXA). This enzyme is encoded by two genes, i.e., *HEXA* and *HEXB*, the products of which are α and β subunits of HEXA, respectively. Mutations in the *HEXA* gene lead to Tay–Sachs disease, while mutations in the *HEXB* gene lead to Sandhoff disease. Both diseases lead to accumulation of GM2-gangliosides, resulting in severe neurodegeneration and neuroinflammation [8]. In order to achieve maximum efficacy of gene therapy, simultaneous delivery of both genes in an equivalent amount is required. We identified two registered AAV clinical trials for GM2-gangliosidoses. Patients in the first study were injected intrathecally with an AAV9-based bicistronic vector [117] containing the HEXB-P2A-HEXA genetic cassette, which consists of *HEXA* and *HEXB* genes separated by a 2A self-cleaving peptide derived from porcine teschovirus-1 [118] (NCT04798235).

In the second study, two AAVrh8-based monocistronic vectors encoding *HEXA* and *HEXB* (AAVrh8-HEXA and AAVrh8-HEXB, respectively) were administered via bilateral thalamic and dual intracisternal magna/intrathecal administration into the cerebrospinal fluid. Prior to clinical trial registration, this strategy has been extensively studied in animal models of GM2-gangliosidosis, the results of which have been reported in a number of scientific publications. For example, several studies have demonstrated the safety and widespread distribution of HEXA in the CNS following intracranial infusion of AAVrh8 encoding *HEXA* and *HEXB* in cats with Sandhoff disease [119,120,121,122]. AAVrh8 is able to cross the BBB and effectively transduce nerve cells and also exhibits reduced tropism for peripheral tissues [123]. Intracranial administration of a mixture of AAVrh8 monocistronic vectors in a mouse model of Sandhoff disease prevented a decrease in the density of neurons in the thalamus [124].

Sandhoff disease cats show some signs of mucopolysaccharidosis, including accumulation of glycosaminoglycans in bones, connective tissue, heart valves, etc., causing the disease to primarily affect these organs (heart valve abnormalities, skeletal changes, and spinal cord compression). Simultaneous intracranial injection of two AAVrh8-encoding feline *HEXA* and *HEXB* genes resulted in reduced accumulation of glycosaminoglycans in the cerebral cortex and liver of experimental cats [125]. The effectiveness of this strategy in skeletal disorders has also been investigated. In cats with Sandhoff disease, accumulation of GM2-gangliosidoses and glycosaminoglycans in chondrocytes causes disruptions in the functions of these cells, along with osteogenesis, which negatively affects the morphology of bones and joints, especially in the metaphyseal and epiphyseal regions of long bones. Gene therapy has been investigated for the treatment of such symptoms and was found to significantly reduce CNS changes associated with Sandhoff disease, including glycosaminoglycans levels in the brain. However, direct injection of the drug into the brain was found to have less of an effect on most peripheral tissues, including the musculoskeletal system [126].

### 3.3. AAV-Mediated Gene Therapy for Fabry Disease

Fabry disease results from mutations in the α-galactosidase A (GLA) gene, leading to enzyme deficiency and accumulation of globotriaosylceramide in cells of various systems. For Fabry disease, six clinical trials of AAVs encoding the *GLA* gene are currently ongoing. AAV6 (NCT04046224 and NCT05039866) and novel modified AAV serotypes have been investigated. One of these novel serotypes is AAVS3, which, judging by the publications of preclinical studies, is most likely a modified version of AAV8 (NCT04040049 and NCT04455230). The second modified AAV serotype has not yet been disclosed by copyright holders; however, the name 4D-C102 (NCT04519749 and NCT05629559) was used in clinical trials. As Fabry disease is a multisystemic disease, vectors in all studies were administered intravenously.

Preclinical studies of AAV6 encoding the human *GLA* gene were conducted in mice with Fabry disease. Intravenous administration of the vector led to a significant increase in GLA activity in plasma and tissues, along with normalization of globotriaosylceramide and lyso-globotriaosylceramide levels in mainly affected areas [61]. For clinical research, an optimized version of this vector was used, which resulted in several times higher levels of GLA activity in plasma and tissues, and the vector also showed a good safety profile (NCT04046224 and NCT05039866).

Intravenous administration of AAV8 (3 × 10^11^ vg/animal) encoding human *GLA* in Fabry mice resulted in a higher level of enzyme production compared to AAV2. At the same time, globotriaosylceramide accumulation was completely prevented in the liver, spleen, heart, kidney, and plasma of animals, and a decrease in peripheral neuropathy was noted [62]. AAV8 is known to be the best candidate for gene delivery to liver cells, with a high efficiency confirmed in many studies in rodents and primates [127]. This serotype has also been found to achieve higher transduction rates of human hepatocytes compared to other AAV serotypes [70].

Capsid 4D-C102 is noted to be specific to muscle tissue. Preclinical studies in mice have shown that 4D-C102 is effective in transducing human cardiomyocytes [128] and has also reduced immunogenicity compared to AAV1 8 and 9 [69].

### 3.4. AAV-Mediated Gene Therapy for Gaucher Disease

Gaucher disease results from mutations in the β-glucocerebrosidase 1 (GBA1) gene, which cause enzyme deficiency, leading to the accumulation of its substrate, glucosylceramide, in macrophages. Gaucher disease is characterized by damage to many internal organs, especially bone marrow, the spleen, and liver, due to infiltration by Gaucher cells (enlarged macrophages containing uncleaved glucocerebroside). Gaucher disease type 1, unlike types 2 and 3, is not associated with neurological impairment [14,129].

Currently, there are four ongoing AAV-based gene therapy clinical trials for Gaucher disease. In all four studies, AAVs were administered intravenously, three using AAV9 encoding the *GBA1* gene (NCT05487599, NCT04411654, and NCT04127578). Results of in vivo studies in a mouse model of Gaucher disease showed that intravenous administration of AAV9 carrying the mouse *GBA1* gene controlled by human cytomegalovirus enhancer and promoter (3 × 10^11^ vg/animal) restores the activity of GBA1 in many organs, prolongs lifespan, and improves neuropathological changes in mice [130].

Mutations in the *GBA1* gene are also associated with risk of Parkinson’s disease [131,132,133]. Intravenous administration of the same vector (AAV9-GBA1) is being investigated in 24 Parkinson’s patients with a mutation in the *GBA1* gene (NCT04127578). In vivo studies support the hypothesis that restoring normal levels of *GBA1* activity can slow the progression of Parkinson’s disease in patients with mutations in the *GBA1* gene [132].

Another clinical study involves the use of a modified AAVS3 capsid, as discussed above for the treatment of Fabry disease (studies are being conducted by the same company; NCT05324943).

### 3.5. AAV-Mediated Gene Therapy for Metachromatic Leukodystrophy

Metachromatic leukodystrophy (MLD) develops as a result of mutations in the arylsulfatase A (*ARSA*) gene. Low levels or a lack of enzyme activity lead to accumulation of sulfatides, mainly in the CNS. MLD is a severe progressive neurodegenerative disease that affects the myelin sheath of cells in the nervous system [21,134].

One of the highly effective AAV serotypes for gene delivery to the nervous system is AAVrh10. In a clinical study, AAVrh10-ARSA was delivered intracerebrally to five patients (NCT01801709). The efficacy and safety of this approach were investigated in MLD mice [71] and nonhuman primates [75]. A single intrastriatal injection of AAVrh10 encoding human gene *ARSA* controlled by CMV/β-actin hybrid promoter led to a decrease in sulfatide levels in the brain and oligodendrocytes. AAVrh10 transduced neurons and oligodendrocytes more efficiently than AAV5, and this vector was also better distributed by axonal transport [71]. Furthermore, a preclinical study of the safety profile and biodistribution of intracerebral administration was conducted in nonhuman primates. No toxicity was reported in the peripheral organs of animals; however, a humoral response to AAVrh10 capsid and the human *ARSA* gene was detected. Administration of a fivefold dose (5.5 × 10^11^ vg/hemisphere) led to an inflammatory process in the brain; however, this effect was absent following the administration of a onefold dose (1.1 × 10^11^ vg/hemisphere). Arylsulfatase A enzymatic activity also exceeded the normal level of endogenous activity by 14–31% following the administration of a onefold dose [75].

### 3.6. AAV-Mediated Gene Therapy for Krabbe Disease

Similar to MLD, Krabbe disease is a progressive demyelinating disease characterized by accumulation of galactosylceramides, predominantly in the nervous system. This disease is caused by decreased activity of galactosylceramidase (GALC) resulting from mutations in its coding gene [135,136,137]. Two AAV serotypes have been studied in clinical trials for Krabbe disease: AAVhu68 and AAVrh10 carrying the *GALC* gene.

In preclinical research, AAVhu68 (a close serotype to AAV9) has been studied in mice, dogs, and nonhuman primates with Krabbe disease. In newborn mice, AAVhu68 encoding codon-optimized human *GALC* controlled by ubiquitous promoter CB7 was injected into the lateral ventricle (5.0 × 10^11^ vg/g brain). According to researchers, AAVhu68 serotype is characterized by broad diffusion and neurotropism after injection into the cerebrospinal fluid. Therefore, the authors showed a significant difference in therapeutic effects achieved by various methods of gene drug delivery. Vector administration into the lateral ventricle led to an increase in lifespan by over 2.5 times compared to intravenous administration of the same number of genomic copies. Dogs received intracisternal magna injections (6.0 × 10^11^ vg/g brain) prior to symptom onset, which resulted in preservation of peripheral nerve myelination and motor function and decreased neuroinflammation and brain demyelination, although some areas remained abnormal. Researchers also investigated the safety of the administration of three doses of the vector into the cisterna magna (5.0 × 10^10^, 1.7 × 10^11^ or 5.0 × 10^11^ vg/g brain) in nonhuman primates. No dose-limiting toxicity was observed [25]. Based on these results, clinical trials of intracisternal magna administration of this vector to children with early infantile Krabbe disease (NCT04771416) were initiated.

In dogs with Krabbe disease, the effect of combined intravenous and intracerebroventricular administration of AAVrh10 encoding *GALC* was studied for both the peripheral and central nervous systems. Dogs were administered AAVrh10 encoding the canine *GALC* gene controlled by CMV-enhancer/chicken β-actin hybrid promoter at two doses (low: 1.2 × 10^12^; high: 3.8 × 10^13^ vg). This total number of vectors was divided equally; one part was administered intravenously, and the other was administered intravenously. A combination of intravenous and intracerebroventricular gene therapies resulted in a clear dose-dependent response and a delay in the onset of clinical signs, as well as increased survival, correction of biochemical defects, and attenuation of neuropathology in animals [78].

Studies on a mouse model of Krabbe disease have shown that the combination of bone marrow transplantation and gene therapy with AAVrh10 significantly increases the lifespan of animals (approximately 10-fold). Mouse models were injected with five doses of AAVrh10 encoding the murine *GALC* gene controlled by human CMV-enhancer/chicken β-actin hybrid promoter at different times after BMT. The dosage had a significant effect on survival in animals. Importantly, delaying transplantation or virus administration resulted in a shortened lifespan of the mice [81]. Based on preclinical data [78,81], a clinical trial of AAVrh10 intravenous administration to patients with Krabbe disease after bone marrow transplantation (NCT04693598) was registered.

## 4. Lentiviruses in Gene Therapy

Transplantation of CD34^+^ genetically modified with LVs encoding therapeutic genes has been widely used in the treatment of various diseases. To date, eight LV-based drugs have been approved by the FDA, six of which (Kymriah, Yescarta, Tecartus, Breyanzi, Abecma, and Carvykti) were developed for CAR T-cell therapy for blood malignancies. The other two drugs (Zynteglo and Skysona), which were developed for the treatment of rare hereditary diseases, were approved in 2022, both of which, in addition to CAR T-therapy drugs, are HSCs (CD34^+^ cells) genetically modified with LVs and have successfully passed clinical trials for the treatment of cerebral adenoleukodystrophy (Skysona, NCT01896102, and NCT03852498) and β-thalassemia (Zynteglo, NCT02906202, and NCT03207009). Regarding sphingolipidoses, 9 out of 10 LV clinical trials registered to date are also aimed at cell-mediated gene therapy (transplantation of CD34^+^ genetically modified LVs), with only one study aimed at intracerebral injection of LV. Moreover, Libmeldy, an autologous CD34^+^ genetically modified by LVs encoding the *ARSA* gene, was approved by the EMA in 2020. Although LV is an integrating type of vector, many studies have established the safety of using third-generation LV.

### 4.1. LV-Mediated Gene Therapy for Fabry Disease

Three LV clinical trials have been registered for Fabry disease, all involving transplantation of CD34^+^ genetically modified with LVs (NCT04999059, NCT03454893, and NCT02800070). The first results of one of these studies were published in 2021 by A. Khan et al. (NCT02800070) [83]. Study results confirmed the robust safety profile of LV-mediated gene therapy. In the study, five male patients with Fabry disease aged 29 to 48 years received transplantation of CD34^+^ genetically modified with LVs encoding GLA in a does of 3.1–13.8 × 10^6^/kg. The number of vector copies in cells varied from 0.68 to 1.43 copies/genome. Following transplantation, a decrease in plasma levels of lyso-globotriaosylceramide was observed in four of five patients, and globotriaosylceramide plasma levels were generally stable in all patients, except for a later increase in two patients. In three patients, IgG antibodies against the enzyme were detected as a result of previously received ERT, which was carried out before gene therapy. Interestingly, the patients showed almost complete elimination or a decrease in IgG antibody titers against the enzyme after gene therapy intervention, without a resurgence in levels, despite continued exposure to the enzyme from the transduced cells. The authors suggested that gene therapy may reduce pre-existing immunity resulting from ERT [83].

### 4.2. LV-Mediated Gene Therapy for Gaucher Disease

Only one LV clinical study has been conducted to date, in which patients with Gaucher disease received genetically modified CD34^+^ transplantation (NCT04145037), with a long-term follow-up study of these subjects ongoing (NCT04836377). For this study, a vector containing a codon-optimized *GBA1* sequence controlled by elongation factor 1α short (EFS) promoter was developed. Preclinical studies of this vector have shown only a small number of integrated LV vectors; however, a good therapeutic effect has been reported in a mouse model. Researchers note that this approach can improve patients’ bone condition, which is a significant outcome, considering that ERT often does not affect the patient’s skeletal system [84]. Results of a case report of the first patient in this clinical study showed a significant reduction in the level of toxic substrate [138].

### 4.3. LV-Mediated Gene Therapy for Metachromatic Leukodystrophy

Using LV in MLD therapy can be considered a successful example of developing a gene therapy approach for sphingolipidoses. Unlike other diseases, there are many published results of gene therapy clinical trials for MLD. Transplantation of LV-modified CD34^+^ has been studied in animals model [85,88,89] and MLD patients [52,87,91], and the results of these studies have been presented in a number of publications by A. Biffi et al. In 2010, the first clinical trials of this approach were initiated (NCT01560182, NCT04283227, and NCT03392987), and after 10 years of productive research, the EMA approved Libmeldy, which contains autologous CD34^+^ cells encoding the *ARSA* gene, for the treatment of MLD. Nonetheless, possible risks of using this drug were noted, among which are possible problems with transplant engraftment, the possibility of malignant transformation due to insertional mutagenesis, the formation of antibodies against ARSA, etc. [139].

The latest published report from a clinical study (NCT01560182) included data from 29 children with presymptomatic or early symptomatic MLD who received transplantation of genetically modified CD34^+^ expressing ARSA at a rate of 4.2–25.9 × 10^6^/kg, with a transduction efficiency of 61–100%.

Two years after transplantation, motor function had improved by more than 10% compared to the control group (untreated natural history cohort of 31 patients). An average increase of 18.7 times in the enzymatic activity of ARSA was also detected in mononuclear cells of peripheral blood compared to the primary level. Moreover, most subjects showed normal cognitive development, along with prevented or delayed demyelination and brain atrophy throughout the follow-up period. The treatment efficacy was particularly evident in patients treated prior to the onset of symptoms. However, a transient increase in anti-ARSA antibodies as a result of gene cell therapy was found in four patients [52].

Other studies have been conducted that included a collective analysis of this approach in patients with both MLD and adrenoleukodystrophy (in the case of adrenoleukodystrophy, LV encoding the ATP-binding cassette subfamily D member 1 (ABCD1) was used) (NCT02559830). A study of intracerebral administration of LV encoding the *ARSA* (NCT03725670) has also been registered. However, no results have been published from preclinical or clinical studies of this approach.

## 5. Conclusions

Although the first officially registered gene therapy clinical trial in 1988 was conducted for a type of sphingolipidosis, active work in this area did not continue until 2014–2016. Although the results of many years of work are available, widespread clinical use of gene therapy methods for sphingolipidoses has not been achieved to date, and these rare severe hereditary diseases remain incurable. Nevertheless, currently available knowledge cannot be overestimated. Gene therapy approaches for sphingolipidoses have developed in two main directions so far, the first of which is based on various AAV serotypes encoding genes for deficient proteins, which are then administered intravenously, intracerebrally, intrathecally, etc., even using combined administration methods in some cases to achieve the best results. The second approach is a combination of gene and cell therapy. In many cases, bone marrow transplantation eased the disease course in sphingolipidoses patients, but levels of endogenous expression of the deficient protein by donor cells were insufficient. As a result of genetic modification, autologous hematopoietic cells began to synthesize a sufficient amount of the therapeutic enzyme. This led to the restoration of sphingolipid metabolism in many affected parts of the body, including the nervous system, as CD34^+^ cells are able to overcome biological barriers and differentiate into microglial cells (resident macrophages of the CNS). Unfortunately, results of many clinical studies are not available yet. Gene therapy methods for sphingolipidoses described in this review are promising; nevertheless, these innovative methods require further research and thorough safety assessments. Another important aspect is the cost of such drugs. The latest approved gene drugs are among the most expensive in the drug market worldwide [140,141]. Therefore, the process of their production requires optimal research and development, as such advanced therapies are not available or affordable for many patients.

## Figures and Tables

**Figure 1 ijms-24-03627-f001:**
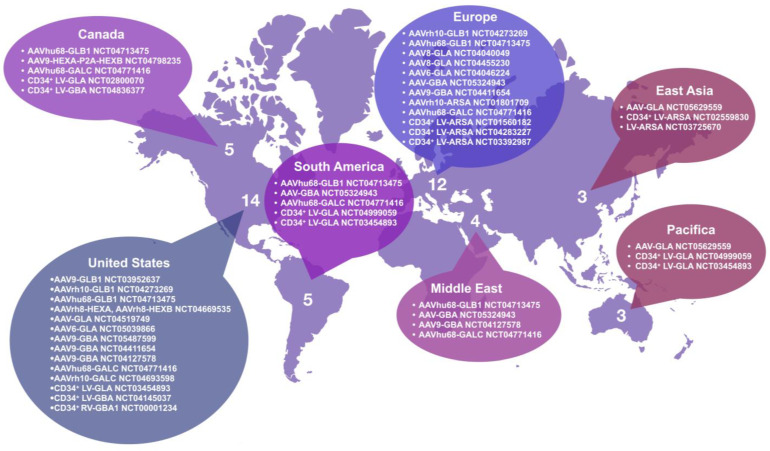
Localization of clinical trials for the treatment of sphingolipidoses. Studies with multiple locations are included in each region.

**Table 1 ijms-24-03627-t001:** Description of sphingolipidoses.

Disease	Inheritance	Mutant Gene	#OMIM	Deficient Protein, UniProt Accession	Description	Estimated Prevalence in General Population	Ref.
GM1-gangliosidosis	AR	*GLB1*	230500 230600 230650	β-Galactosidase 1, P16278	Predominant damage to the nervous system, various degrees of neurodegeneration, and skeletal anomalies	1:100,000– 200,000	[7]
GM2-gangliosidoses, Tay–Sachs disease	AR	*HEXA*	272800	β-Hexosaminidase A, α-subunit P06865	Predominant damage to the nervous system and various degrees of neurodegeneration	1:100,000	[8]
GM2-gangliosidoses, Sandhoff disease	AR	*HEXB*	268800	β-Hexosaminidase A, β-subunit P07686		1:500,000	[9,10,11]
AB variant of GM2-gangliosidosis	AR	*GM2A*	272750	GM2A, P17900		Extremely rare	[8,12]
Fabry disease	XL	*GLA*	301500	α-Galactosidase A, P06280	Multisystemic disease; progressive renal failure; skin, cardiovascular, and nervous system lesions	1:100,000– 500,000	[13]
Gaucher disease	AR	*GBA1*	230800 230900 231000 231005 608013	β-Glucocerebrosidase 1, P04062	Multisystemic disease, predominant damage to cells of mononuclear phagocyte origin (Gaucher cells); varying degree of damage to the nervous system; high risk of developing Parkinson’s disease	1:40,000– 60,000	[14,15,16]
SapC deficiency, atypical Gaucher disease	AR	*PSAP*	610539	Saposin C, P07602		Extremely rare	[17,18,19,20]
Metachromatic leukodystrophy	AR	*ARSA*	250100	Arylsulfatase A, P15289	Damage to the white matter or myelin sheath in nervous system cells	1:40,000– 160,000	[21]
SapB deficiency, atypical metachromatic leukodystrophy	AR	*PSAP*	249900	Saposin B, P07602		Extremely rare	[22,23,24]
Krabbe disease	AR	*GALC*	245200	Galactocerebrosidase, P54803	Damage to the white matter or myelin sheath in nervous system cells	1:100,000	[25,26,27,28]
SapA deficiency, atypical Krabbe disease	AR	*PSAP*	611722	Saposin A, P07602		Extremely rare	[29,30]
Type A and B Niemann–Pick disease	AR	*SMPD1*	257200 607616	Sphingomyelinase, Q9NY59	Damage to the nervous system, bone marrow, spleen, and, in some cases, lungs	1:250,000	[31,32,33]
Farber disease	AR	*ASAH1*	228000	Acid ceramidase, Q13510	Multisystem disease with progressive joint deformation and dysfunctions of multiple organ systems, including the nervous system	<1:1,000,000	[34,35,36]
Combined saposin deficiency	AR	*PSAP*	611721	Saposins A, B, C, and D, P07602	Fatal disorder affecting the nervous system	Extremely rare	[22,37]

**Table 2 ijms-24-03627-t002:** Clinical studies of gene therapy methods for the treatment of sphingolipidoses.

Disease	Vector	Gene	Method of Administration	Participants	Official Title	Clinical Study ID	Years	Links
Adeno-associated viral vectors								
GM1-gangliosidosis	AAV9	*GLB1*	Intravenous	45	A Phase 1/2 Study of Intravenous Gene Transfer with an AAV9 Vector Expressing Human Beta-galactosidase in Type I and Type II GM1 Gangliosidosis	NCT03952637	2019–2027	[66]
	AAVrh10	*GLB1*	Intracisternal	16	An Open-Label Adaptive-Design Study of Intracisternal Adenoassociated Viral Vector Serotype rh.10 Carrying the Human β-Galactosidase cDNA for Treatment of GM1 Gangliosidosis	NCT04273269	2021–2030	[67]
	AAVhu68	*GLB1*	Intracisternal magna	20	Phase 1/2 Open-Label, Multicenter Study to Assess the Safety, Tolerability and Efficacy of a Single-Dose of PBGM01 Delivered into the Cisterna Magna of Subjects with Type 1 (Early Onset), and Type 2a (Late Onset) Infantile GM1 Gangliosidosis	NCT04713475	2021–2029	[68]
GM2-gangliosidosis	AAV9	*HEXA*-*P2A*-*HEXB*	Intrathecal	3	Phase 1/2, Open-Label Clinical Study to Evaluate the Safety and Efficacy of Intrathecal TSHA-101 Gene Therapy for Treatment of Infantile Onset GM2 Gangliosidosis	NCT04798235	2021–2027	
	AAVrh8	*HEXA*, *HEXB*	Bilateral thalamic, intrathecal	18	A Two-Stage, Dose-Escalation and Safety & Efficacy Study of Bilateral Intraparenchymal Thalamic and Intracisternal/Intrathecal Administration of AXO-AAV-GM2 in Tay–Sachs or Sandhoff Disease	NCT04669535	2021–2028	
Fabry disease	AAV	*GLA*	Intravenous	18	An Open-label, Phase 1/2 Trial of Gene Therapy 4D-310 in Adults with Fabry Disease	NCT04519749	2020–2027	[69]
					An Open-Label, Phase 1/2a Trial of Gene Therapy 4D-310 in Adults with Fabry Disease and Cardiac Involvement	NCT05629559	2022–2028	
	AAV8	*GLA*	Intravenous	15	A Phase 1/2, Baseline-controlled, Non-randomized, Open-label, Single-ascending Dose Study of a Novel Adeno-associated Viral Vector (FLT190) in Patients With Fabry	NCT04040049	2019–2022	[70]
				50	A Multicenter, Long-term, Follow-up Study to Investigate the Safety and Durability of Response Following Dosing of an Adeno-associated Viral Vector (FLT190) in Subjects With Fabry Disease	NCT04455230	2020–2026	
	AAV6	*GLA*	Intravenous	48	A Phase I/II, Multicenter, Open-Label, Single-Dose, Dose-Ranging Study to Assess the Safety and Tolerability of ST-920, an AAV2/6 Human Alpha Galactosidase A Gene Therapy, in Subjects with Fabry Disease	NCT04046224	2019–2024	[61]
					Long-Term Follow-up of Fabry Disease Subjects Who Were Treated With ST-920, an AAV2/6 Human Alpha Galactosidase A Gene Therapy	NCT05039866	2021–2029	
Gaucher disease	AAV	*GBA*	Intravenous	18	A Phase 1/2, Open-label, Safety, Tolerability, and Efficacy Study of FLT201 in Adult Patients with Gaucher Disease Type 1 (GALILEO-1)	NCT05324943	2022–2025	
	AAV9	*GBA*	Intravenous	15	An Open-label, Dose-Finding, Phase 1/2 Study to Evaluate the Safety and Tolerability of a Single Intravenous Dose of LY3884961 in Patients with Peripheral Manifestations of Gaucher Disease (PROCEED)	NCT05487599	2022–2030	
				15	An Open-label, Phase 1/2 Study to Evaluate the Safety and Efficacy of Single-dose LY3884961 in Infants with Type 2 Gaucher Disease	NCT04411654	2021–2028	
				24	A Phase 1/2a Open-Label Ascending Dose Study to Evaluate the Safety and Effects of LY3884961 in Patients with Parkinson’s Disease With At least One GBA1 Mutation (PROPEL)	NCT04127578	2020–2028	
Metachromatic leukodystrophy	AAVrh10	*ARSA*	Intracerebral	5	A Phase I/II, Open Labeled, Monocentric Study of Direct Intracranial Administration of a Replication Deficient Adeno-associated Virus Gene Transfer Vector Serotype rh.10 Expressing the Human ARSA cDNA to Children with Metachromatic Leukodystrophy	NCT01801709	2014–2029	[71,72,73,74,75]
Krabbe disease	AAVhu68	*GALC*	Intracisternal magna	24	A Phase 1/2 Open-Label, Multicenter Dose-Ranging and Confirmatory Study to Assess the Safety, Tolerability and Efficacy of PBKR03 Administered to Pediatric Subjects with Early Infantile Krabbe Disease (Globoid Cell Leukodystrophy)	NCT04771416	2022–2030	[25]
	AAVrh10	*GALC*	Intravenous	6	A Phase 1/2 Clinical Study of Intravenous Gene Transfer with an AAVrh10 Vector Expressing GALC in Krabbe Subjects Receiving Hematopoietic Stem Cell Transplantation (RESKUE)	NCT04693598	2021–2024	[76,77,78,79,80,81,82]
Lentiviral vectors								
Fabry disease	LV	*GLA*	Transplantation of CD34^+^ genetically modified with LVs	9	Long-Term Follow-up Study of Subjects with Fabry Disease Who Received Lentiviral Gene Therapy in Study AVRO-RD-01-201	NCT04999059	2019–2036	
				11	An Open-Label, Multinational Study of The Efficacy and Safety of Ex Vivo, Lentiviral Vector-Mediated Gene Therapy AVR-RD-01 For Treatment-Naive Subjects with Classic Fabry Disease	NCT03454893	2018–2022	
	LV	*GLA*	Transplantation of CD34^+^ genetically modified with LVs		Clinical Pilot Study of Autologous Stem Cell Transplantation of Cluster of Differentiation 34 Positive (CD34^+^) Cells Engineered to Express Alpha Galactosidase A in Patients With Fabry Disease	NCT02800070	2016–2024	[83]
Gaucher disease	LV	*GBA*	Transplantation of CD34^+^ genetically modified with LVs	16	The Guard1 Trial, an Open-Label, Multinational Phase 1/2 Study of the Safety and Efficacy of Ex Vivo, Lentiviral Vector-Mediated Gene Therapy AVR-RD-02 for Subjects with Type 1 Gaucher Disease	NCT04145037	2019–2023	[84]
					A Long-Term Follow-up Study of Subjects with Gaucher Disease Who Previously Received AVR-RD-02	NCT04836377	2021–2038	
Metachromatic leukodystrophy	LV	*ARSA*	Transplantation of CD34^+^ genetically modified with LVs	20	Gene Therapy for Metachromatic Leukodystrophy	NCT01560182	2010–2025	[52,85,86,87,88,89,90,91]
				6	An Open-Label, Non-randomized Trial to Evaluate the Safety and Efficacy of a Single Infusion of OTL-200 in Patients With Late Juvenile (LJ) Metachromatic Leukodystrophy (MLD)	NCT04283227	2022–2031	
				10	A Single Arm, Open-Label, Clinical Study of Cryopreserved Autologous CD34^+^ Cells Transduced with Lentiviral Vector Containing Human ARSA cDNA (OTL-200), for the Treatment of Early Onset Metachromatic Leukodystrophy (MLD)	NCT03392987	2018–2028	
	LV	*ARSA*	Transplantation of CD34^+^ genetically modified with LVs	50	A Phase I/II Clinical Trial of Lentiviral Hematopoietic Stem Cell Gene Therapy for Treatment of Developed Metachromatic Leukodystrophy and Adrenoleukodystrophy	NCT02559830	2015–2025	[92,93,94,95]
	LV	*ARSA*	Intracerebral	10	Gene Therapy for Metachromatic Leukodystrophy (MLD) Using a Self-inactivating Lentiviral Vector (TYF-ARSA)	NCT03725670	2018–2020	
Retroviral vectors								
Gaucher disease	Retrovirus	*GBA1*	Transplantation of CD34^+^ genetically modified with a retrovirus	120	Retroviral-Mediated Transfer and Expression of Glucocerebrosidase and Ceramidtrihexosidase (a-Galactosidase A) cDNA’s in Human Hematopoietic Progenitor Cells	NCT00001234	1988–2002	[96,97,98]

## Data Availability

Not applicable.

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
