# Peer review of "Gene Therapy of Sphingolipid Metabolic Disorders"

_ijms, 2023, doi:10.3390/ijms24043627_

Round 1
Reviewer 1 Report
In this review, Shaimardanova et al. comprehensively summarized gene therapies for sphingolipidoses. The authors broke down each of the 10 genetic disorders and discussed gene treatments in detail. They also highlighted AAVs and stem cell therapies to be the most promising for rectifying sphingolipid metabolic disorders.
The review is well-written, and I just have two comments:
1) At the beginning of the introduction, the authors could provide more background for lysosomal storage disorders using the following comprehensive reviews: PMID: 30275469, PMID: 32032518, and recent scientific discoveries: PMID: 36131016, PMID: 36161949, bioRxiv: 517422.
2) The authors could discuss and compare other non-gene therapies for sphingolipidoses in the discussion.
Minor points:
The full name of FDA is usually referred to as the U.S. Food and Drug Administration
For Reference 127, it looks like some information was missing (accessed on)
Beyond these minor critiques, I would recommend acceptance after the corrections are made as mentioned above.
Author Response
We thank the reviewer for the valuable comments. We have improved the manuscript according to the suggestions.
1) More information on lysosomal storage disorders was added, as recommended, on page 1, lines 23-29
2) A discussion of other non-gene therapies for sphingolipidoses was added on page 3, lines 50-75)
3) Minor points were revised as recommended.
Reviewer 2 Report
I believe the paper “gene therapy of sphingolipid metabolic disorders”
Authors: Alisa A. Shaimardanova, Valeriya V. Solovyeva, Shaza S. Issa, Albert
A. Rizvanov
is worth publishing provided that the authors develop certain aspects
Introduction
In the introduction, the authors briefly overview the currently available therapeutic approaches for sphingolipidosis. However, they fail to discuss the most recent approaches.
For example, the regulation of proteostasis has recently emerged as a very promising entry point for treating lysosomal storage disorders and, therefore, also for sphingolipidoses. It has many advantages, like the possibility of drug repositioning, and the use of small molecules capable of overcoming the blood-brain barrier. Consequently, it would be appropriate to discuss this approach. Please, have a look at the following papers:
10.1016/j.ebiom.2018.11.037
10.3390/ijms23095105
10.1093/hmg/dds145
10.1021/cb100321m
Also, translational readthrough is a very innovative approach. Please, have a look here:
10.1080/15476286.2019.1676115
10.1371/journal.pone.0135873
Results
The authors should explicitly say how they generate the list of diseases used to build for Table 1; furthermore, it would be valuable to report uniprot accession for each protein mentioned in the table.
Also, is there any reason behind the absence of some diseases from clinical trial databases? Are those diseases also less studied in general? Maybe rarer? Finally, can the authors to briefly quantify the number of methods falling into the two approaches and how such a number changes in time (e.g., if both are increasing).
References
The vast majority of the bibliography is three or fewer years old, as it should be in a review.
Author Response
We thank the reviewer for the valuable comments. We have improved the manuscript according to the suggestions.
1) A discussion of the two novel approaches; regulation of proteostasis and translational readthrough, was added on page 3, lines 59-71
2) Diseases in Table 1 were listed as following: sphingolipidoses that are grouped according to a certain phenotype or protein deficiency, were listed together. UniProt accession for each protein was added in the Table 1
3) The probable reason for the absence of certain diseases in the databases of clinical trials was indicated (Page 4, Lines 116-120)
4) An explanation of the trend regarding number of registered clinical trials of LV and AAV was added, as suggested (Page 4, Lines 120-124).
Round 2
Reviewer 2 Report
The paper was improved but some of the suggestions were not taken into account
In the introduction, the authors briefly overview the currently available therapeutic approaches for sphingolipidosis. However, they fail to discuss the most recent approaches.
For example, the regulation of proteostasis has recently emerged as a very promising entry point for treating lysosomal storage disorders and, therefore, also for sphingolipidoses. It has many advantages, like the possibility of drug repositioning, and the use of small molecules capable of overcoming the blood-brain barrier. Consequently, it would be appropriate to discuss this approach. Please, have a look at the following papers:
10.1016/j.ebiom.2018.11.037
10.3390/ijms23095105
10.1093/hmg/dds145
10.1021/cb100321m
Also, translational readthrough is a very innovative approach. Please, have a look here:
10.1080/15476286.2019.1676115
10.1371/journal.pone.0135873
Author Response
Dear Reviewer,
References to publications were not added in the previous version of the manuscript by mistake, I apologise for this. References added in corrected version.
Best wishes,
Alisa Shaimardanova